# Effects of Nanosecond Pulsed Electric Field (nsPEF) on a Multicellular Spheroid Tumor Model: Influence of Pulse Duration, Pulse Repetition Rate, Absorbed Energy, and Temperature

**DOI:** 10.3390/ijms241914999

**Published:** 2023-10-08

**Authors:** Rosa Orlacchio, Jelena Kolosnjaj-Tabi, Nicolas Mattei, Philippe Lévêque, Marie Pierre Rols, Delia Arnaud-Cormos, Muriel Golzio

**Affiliations:** 1University Bordeaux, CNRS, IMS, UMR 5218, 33400 Talence, France; rosa.orlacchio@u-bordeaux.fr; 2École Pratique des Hautes Études, PSL Research University, 75014 Paris, France; 3Institut de Pharmacologie et de Biologie Structurale (IPBS), Université de Toulouse, CNRS, Université Toulouse III—Paul Sabatier (UT3), 31062 Toulouse, France; jelena.kolosnjaj-tabi@ipbs.fr (J.K.-T.); nicolas.mattei@ipbs.fr (N.M.); marie-pierre.rols@ipbs.fr (M.P.R.); 4University Limoges, CNRS, XLIM, UMR 7252, 87000 Limoges, France; philippe.leveque@unilim.fr (P.L.); delia.arnaud-cormos@unilim.fr (D.A.-C.); 5Institut Universitaire de France (IUF), 75005 Paris, France

**Keywords:** buffer conductivity, cancer therapies, electropulsation, hyperthermia, multicellular cancer spheroids, nanosecond pulsed electric field (nsPEF), pulse repetition rate

## Abstract

Cellular response upon nsPEF exposure depends on different parameters, such as pulse number and duration, the intensity of the electric field, pulse repetition rate (PRR), pulsing buffer composition, absorbed energy, and local temperature increase. Therefore, a deep insight into the impact of such parameters on cellular response is paramount to adaptively optimize nsPEF treatment. Herein, we examined the effects of nsPEF ≤ 10 ns on long-term cellular viability and growth as a function of pulse duration (2–10 ns), PRR (20 and 200 Hz), cumulative time duration (1–5 µs), and absorbed electrical energy density (up to 81 mJ/mm^3^ in sucrose-containing low-conductivity buffer and up to 700 mJ/mm^3^ in high-conductivity HBSS buffer). Our results show that the effectiveness of nsPEFs in ablating 3D-grown cancer cells depends on the medium to which the cells are exposed and the PRR. When a medium with low-conductivity is used, the pulses do not result in cell ablation. Conversely, when the same pulse parameters are applied in a high-conductivity HBSS buffer and high PRRs are applied, the local temperature rises and yields either cell sensitization to nsPEFs or thermal damage.

## 1. Introduction

Nanosecond pulsed electric fields (nsPEF) are emerging as a cancer treatment modality. High-intensity (tens of MV/m) pulses of short duration (hundreds of ns) showed excellent capabilities to induce cellular death of cancerous cells within in vitro [1,2,3] and in vivo [4,5] models, as well as in humans, as reported in human trials [6,7]. In addition, the unique capability of nsPEF to bypass the cell plasma membrane may elicit specific intracellular responses, including the activation of regulated cell death and the initiation of an adaptive immune response [8,9] showing high efficacy in different in vivo tumor models, including murine hepatocellular carcinoma [10], melanoma [11,12,13,14], or breast cancer [15,16]. Compared to other cancer ablation modalities, nsPEF exposure allows direct and localized tumor ablation, minimizing the damage to healthy tissues surrounding the tumor, avoiding the use of chemotherapeutic drugs, and avoiding associated adverse reactions. Moreover, recent studies suggest that nsPEF ablation also results in reduced neuromuscular stimulation, which is responsible for pain and muscle contraction, which is one of the major drawbacks in patients exposed to PEFs with a duration between 100 µs and 1 ms [17]. The considerable progress in pulsed power nowadays allows the generation and delivery of high-intensity ultrashort pulses of the order of a few tens of ns or even picoseconds (psPEF) [18,19], paving the way toward novel precise cancer therapies [4]. In particular, under specific exposure conditions, short pulses between 2 and 10 ns may induce a plethora of bioeffects, including, for instance, disruption or modulation of microtubule dynamics [20], plasma membrane depolarization via voltage-gated K^+^, Ca^2+^, and transient receptor potential ion channel subfamily M, member 8 (TRPM8) channels [21], stimulation of catecholamine release in chromaffin cells [22], membrane permeabilization [23], phosphatidylserine translocation in lipid bilayers [24], or enhancement of cellular differentiation [25].

We have recently demonstrated that trains of at least 100 unipolar electric pulses of 10 ns (50 kV/cm, delivered at a repetition rate of 20 Hz) can trigger cellular permeabilization and mortality in 3D multicellular spheroids derived from HCT-116 colorectal carcinoma cells [26]. These results suggest that ultrashort pulses with a duration inferior to 100 ns are also excellent candidates for non-invasive and selective targeting of tumors. In our previous study [26], we explored the mortality of 3D-grown cancer cells up to 6 days following exposure to nsPEF, and we showed that 10 ns pulses can induce cell death in 3D cancer cell models. Multicellular cancer cell spheroids are structurally closer to small avascular solid tumors and better represent the complex tissue environment compared to adherent cells, suggesting that the observed effects are likely to occur in vivo.

In this study, we performed further research focusing on the effect of different pulse parameters of duration equal to or shorter than 10 ns. Namely, we assessed the cellular viability and growth of 3D-cultured cancer cells as a function of (1) pulse duration (2–10 ns), (2) pulse repetition rate (PRR) (20 and 200 Hz), (3) cumulative time duration (1–5 µs), and (4) absorbed electrical energy density up to 81 mJ/mm^3^ in saccharose-containing low-conductivity buffer (0.2 S/m), named “ZAP”, and up to 700 mJ/mm^3^ in high-conductivity buffer (1.4 S/m), the HBSS. The nsPEF-induced temperature elevation in the pulsing buffers was studied as well. Live cell imaging was used to investigate the cellular viability and evolution of cell growth in 3D multicellular spheroids made with a human colon cancer-derived cell line (HCT-116), constitutively expressing the green fluorescent protein (GFP).

## 2. Results

### 2.1. Effects of Pulse Durations in HBSS Buffer at 20 Hz

Live fluorescence microscopy was used to assess the long-term impact of nsPEF exposure by observing (1) the induced cell death in the spheroids 32 h after nsPEF exposure and (2) cells’ ability to re-grow over a five-day period. Spheroids were subjected to 500 unipolar pulses of 2, 4, 6, 8, and 10 ns duration (referred to as Protocol 1), applied at a 20 Hz repetition rate. Ten minutes after nsPEF exposure, 1 µM PI was added to the culture medium. At this point, any pulse-induced membrane resealing was considered to have occurred, and PI was used to evidence non-viable cells. Cells in spheroids exposed to 2 or 4 ns pulses’ durations did not uptake the PI. Pulses lasting 6 ns induced a weak uptake of PI within the spheroid’s peripheral cells, observed one day after nsPEF exposure. This effect was more pronounced following exposure to 8 ns and 10 ns pulses, where the spheroids’ external rim detachment was observed, most likely due to a loss of cohesion of the structure (Figure 1A). For all conditions, the mean fluorescence intensity measurements showed that PI uptake occurred during the first 8 h following exposure and then remained constant (Figure 1B).

Spheroid growth curves (GFP fluorescence area of viable cells) showed that spheroids treated with 2, 4, and 6 ns pulse durations had comparable spheroid sizes and grew at a similar rate as sham-exposed control spheroids. Spheroids exposed to 8 ns pulse duration were affected by the pulses and exhibited a smaller GFP area, corresponding to a smaller number of cells that remained viable after nsPEF exposure, but the slope of the growth curve (corresponding to spheroids growth rate) was comparable to the control and cells exposed to 2, 4, and 6 ns pulses, indicating that the cell death observed with PI during the first 24 h did not affect spheroids growth. Spheroids exposed to 10 ns pulse duration exhibited an almost negligible GFP area immediately after treatment and were characterized by slower growth than control spheroids during the 5-day follow-up.

Taken together, the results obtained with the protocol 1 showed that the increase in the pulse duration from 2 to 10 ns under these specific exposure conditions, allowed an increase in cell death and a decrease in spheroid growth.

### 2.2. Effects of Pulse Durations in HBSS Buffer at 200 Hz

Similar experiments as described in the previous section were performed, but the PRR was increased to 200 Hz. Spheroids were subjected to 500 unipolar pulses at 200 Hz of 2, 6, and 10 ns duration (referred to as Protocol 2), and 10 min following nsPEF exposure, 1 µM PI was added to the culture medium. All cells in the spheroids were viable following exposure to a 2 ns pulse duration, with no cells showing PI uptake. Following 6 and 10 ns pulse durations, cells within the entire spheroid were non-viable, as PI uptake was observed throughout the spheroid. The spheroid did not exhibit cell detachment (Figure 2A), and the external rim remained cohesive. Spheroid growth curves (GFP fluorescence area of viable cells) showed that spheroids treated with a 2 ns pulse duration grew at a similar rate as sham-exposed control spheroids. Spheroids exposed to 6 and 10 ns pulse durations did not grow during the 5-day follow-up (Figure 2B). Taking into account the temperature increases upon 6 ns and 10 ns pulses application in Protocol 2 (Table 1), the negative cellular effects observed in this protocol are most likely due to the local temperature increase upon pulses application in the HBSS buffer.

### 2.3. Effects of Pulse Durations with Constant Cumulative Time Duration and Absorbed Energy Density in HBSS Buffer at 20 Hz and 200 Hz

Increasing pulse duration while keeping the number of pulses constant led to an increase in the cumulative time duration from 1 to 5 μs and an increase in the absorbed energy density from 140 to 700 m J/mm^3^ in the high-conductivity buffer. Therefore, we decided to evaluate the role of the pulse duration while keeping both the cumulative time duration and the absorbed electrical energy density constant at the maximum values of 5 μs and 700 mJ/mm^3^, respectively. To do so, spheroids were subjected to 2500, 833, and 500 unipolar pulses lasting 2, 6, and 10 ns, respectively, delivered at a PRR of 20 Hz (referred to as Protocol 3). Following 2, 6, and 10 ns pulse durations, we observed PI uptake into the spheroid’s peripheral cells at 24 h post-exposure, which was (i) proportional to pulse duration with PI penetration into the external rim for 2 ns pulses, (ii) almost complete PI penetration for 6 ns pulses, and (iii) complete penetration for 10 ns pulses. The rim of exposed spheroids started to detach, and the spheroids’ surfaces were rich in membrane blebs and cellular debris (Figure 3A). The spheroid growth curves (GFP fluorescence area of viable cells) showed that spheroids with 2, 6, and 10 ns pulse durations exhibited a smaller viable cell fraction than control during the 5-day follow-up (Figure 3B). Note that Figure 3A shows representative micrographs of spheroids at the timepoint of 24 h post-pulse exposure. At this moment, the GFP fluorescence was too low to be detected by fluorescence microscopy (e.g., only a small number of cells survived and did not generate enough detectable signal). Conversely, Figure 3B traces the GFP over a period of about 130 h and thus captures the longitudinal evolution of the increase in GFP. While the GFP cells were not sufficiently fluorescent to be visible in the micrograph at 24 h, over time, the cells that survived created an increasingly bigger GFP zone that is reflected on the curve over time. Taking into account the measured temperature increase (Table 1), we assume that these effects can be principally attributed to PEF and not to hyperthermia.

Experiments were also performed at 200 Hz, referred to as Protocol 4, and spheroids were subjected to 2500, 833, and 500 unipolar pulses lasting 2, 6, and 10 ns, respectively. As for the other protocols, 1 µM PI was added to the culture medium. Following 2, 6, and 10 ns pulse durations, cells within the center of the spheroid were non-viable, and PI uptake was observed throughout the entire spheroid. No disruption, rim detachment, or cellular blebs were observed (Figure 4A). Spheroid growth curves (GFP fluorescence area of viable cells) showed that spheroids exposed to 2, 6, and 10 ns pulse durations did not grow during the 5-day follow-up (Figure 4B). Results obtained with Protocol 4 showed that the increase in PRR while keeping the cumulative time duration and absorbed energy density constant induced cell death in all cells constituting the spheroid, regardless of the pulse duration. Even 2 ns pulses could induce cell mortality. Considering the temperature elevation as detailed in Table 1 and the lack of morphological signs of electroporation-induced cell death (characterized by membrane blebs and important loss of cell cohesion), we stipulate that the cell death induced by Protocol 4 is related to hyperthermia, including when the pulse duration was as low as 2 ns.

### 2.4. Effect of Low-Conductivity Sucrose-Containing (ZAP) Buffer

Pulsing buffer nature, and in particular its conductivity, may impact the temperature elevation during nsPEF exposure. To explore the effect of PEF in different pulsing buffers, the same experiments reported in the previous sections were performed using pulse Protocols 1, 2, 3, and 4, applied in a low-conductivity sucrose-containing medium (ZAP). In ZAP buffer, the temperature increase (ΔT) upon nsPEF exposure was minimal (e.g., the highest temperature elevation was approximately 4.4 °C). Under all exposure conditions, all cells in the spheroids were viable and grew at a similar rate as sham-exposed control spheroids (Figure 5). These experiments suggest that temperature could have an important, if not major, role in reduced cell viability and a subsequent decrease in spheroids’ growth.

### 2.5. Induced Temperature Elevation

#### 2.5.1. Electro-Induced Temperature Elevation

Peak temperatures recorded at the end of the pulse train for the four protocols are summarized in Table 1. Results are reported as the mean ± standard deviation for at least three measurements per condition.

The maximal temperature elevation was recorded after exposure to Protocol 4 with a 100 kV/cm electric field amplitude in HBSS buffer (Table 1 and Figure 6A). The change in temperature (ΔT) corresponds to the temperature increase obtained at the end of the nsPEF exposure. Specifically, the temperature increase (ΔT) at the end of the exposure was equal to 22.0 °C, 42.8 °C, and 46.8 °C for 2, 6, and 10 ns trains, respectively. Conversely, in ZAP buffer (Table 1), the heating under the same exposure protocol at 90 kV/cm increased by approximately 3.2 °C, 3.7 °C, and 4.4 °C for 2, 6, and 10 ns trains, respectively, and nsPEF exposure did not affect the spheroids viability and growth (Figure 5).

A representative example of the peak temperatures for the harshest exposure condition when 500 pulses were applied for every pulse duration (PRR 200 Hz, Protocol 2) in HBSS is presented in Figure 6B. At the end of the exposure, the maximum temperature increase (ΔT) varied approximatively between 12.6 °C and 46.8 °C when using 2 and 10 ns unipolar pulses, respectively. In contrast, under Protocol 2 applied to spheroids in ZAP buffer, peak temperature only increased for 1.2 °C and 4.4 °C at the end of the pulse trains of 2 and 10 ns duration pulses, respectively.

#### 2.5.2. Hyperthermia by Convection

In order to assess the sole effect of increased temperature on spheroids viability and growth on HCT-116 multicellular spheroids without concomitant exposure to a pulsed electric field, we exposed the spheroids to different temperatures (50 °C, 60 °C, 70 °C, or 75 °C, respectively), and PI internalization and GFP fluorescence were used to assess the long-term impact of hyperthermia on spheroids morphology and behavior (viability and growth). Spheroids morphology, as well as GFP expression (green fluorescence) and PI internalization (red fluorescence indicating dead cells), were followed over a period of 5 days. The characteristic micrographs obtained one day post-exposure (Figure 7A) and the corresponding spheroids growth curves, expressed as GFP positive area or total area (Figure 7B,C), and induced cell death, characterized by PI fluorescence (Figure 7D), indicate that spheroids were affected by hyperthermia at 50 °C and above.

## 3. Discussion

Nanosecond PEF exposure represents a promising cancer treatment modality that has been experimentally explored during the last 20 years, with the first successful clinical trials recently performed on basal cell carcinoma using electric pulses of 100–200 ns and field amplitudes of the order of 30 kV/cm [6,7]. Compared to other cancer treatment modalities, high-intensity short electric pulses can easily penetrate the cell plasma membrane and, under specific exposure conditions, possibly target the mitochondria and the endoplasmic reticulum [27]. This initiates a cascade of intracellular events, including an increase in the production of reactive oxygen species (ROS) or the release of cytochrome c and intracellular Ca^2+^, leading to drug-free regulated cell death [9]. Most of the studies, recently discussed in detail in [28], have explored the potential bioeffects of electric pulses with pulse duration ≥ 10 ns, while little is known about the effects of shorter (<10 ns) pulses. In addition, the effects of nsPEF were rarely studied on realistic three-dimensional in vitro cancer models, such as multicellular spheroids. Shortening pulse duration is also very attractive to possibly reach deep-seated tumors using contactless technologies [29]. Indeed, when the pulse width is reduced, the dielectric properties of cells become important, paving the way toward propagative delivery systems that have the considerable advantage of eliminating the need for surgery to insert the electrodes [29].

In this study, to further explore the potential effects of ultrashort pulses, we have investigated the impact of nsPEF with durations between 2 and 10 ns and an electric field strength of about 100 kV/cm on cellular viability and growth of multicellular spheroids under different exposure protocols.

Overall, our results show that in a low-conductivity sucrose-containing ZAP buffer, nsPEF does not modify cell viability or long-term spheroid growth, as indicated by the experiments performed (Figure 5). The absorbed energy density delivered during our experimental protocols in ZAP buffer ranges from 16 to 81 mJ/mm^3^ with a maximum temperature elevation (ΔT) of only around 4.4 °C, which was reached following the highest exposure energy. Interestingly, despite these values being within the similar range of the absorbed energy density used in standard protocols [30], such as electrochemotherapy (ECT, 8 pulses, 100 µs, 1000 V/cm, u_HBSS_ = 11 mJ/mm^3^ and u_ZAP_ = 1.6 mJ/mm^3^) or irreversible electroporation (IRE, 80 pulses, 100 µs, 2000 V/cm, [30], u_HBSS_ = 441 mJ/mm^3^ and u_ZAP_ = 64 mJ/mm^3^), the effects of nsPEF on multicellular spheroids differ, most likely due to different interaction mechanisms of long versus ultra-short pulses.

However, when experiments were conducted in HBSS, the high-conductivity buffer, both spheroids’ growth and cellular viability were affected. In HBSS, the absorbed energy density delivered during our experimental protocols varied from 140 to 700 mJ/mm^3^ for the lowest and highest pulse width, respectively, with a maximum temperature elevation (ΔT) of approximately 46 °C. When multicellular spheroids were exposed in HBSS to 500 pulses at 20 Hz (Protocol 1), the induced bioeffects were mainly dependent on pulse duration. Indeed, PI uptake and reduction of cellular growth were observed only with pulses > 6 ns. Mild ΔT of about 9, 11, and 17 °C were recorded following 6, 8, and 10 nsPEF trains, respectively. The maximum temperature of the exposed medium depends on the ambient temperature, and in our setup, we exposed the media at room temperature of about 23 ± 1 °C. This resulted in an exposed bulk temperature of approximately 32, 33, and 40 °C. When we exposed the spheroids under Protocol 3, 2500 pulses of 2 ns and 833 pulses of 6 ns at 20 Hz, cellular death was induced by pulses with length ≥ 2 ns, suggesting that the number of pulses plays an important role in defining the amplitude of the effect. For these experiments, the bulk temperature remained at or below 41 °C for very short time lapses (up to a few tens of seconds). Here it is important to note that hyperthermia is generally defined as long-term 40–41 °C exposure for 6–72 h (also termed “low-temperature hyperthermia”), while exposure to 42–45 °C for 15–60 min is considered as “moderate-temperature hyperthermia” [31]. Thermal ablation, or “high-temperature hyperthermia”, occurs at temperatures > 50 °C, and is applied for periods higher than 4–6 min [31].

Under our experimental conditions used in Protocols 1 and 3 in HBSS buffer (500 to 2500 pulses, lasting 2 to 10 ns, applied at a PRR of 20 Hz with an electric field intensity of 100 kV/cm), the mild temperature increase clearly enhanced nsPEF effects. This finding is in line with the findings highlighted in several studies [32,33,34,35,36]. For example, Camp et al. [34] demonstrated that a temperature level above physiological conditions (>37 °C) during exposure to 200 ps pulses strongly boosted the induction of cellular death of liver cancer cells using an energy density below 60 mJ/mm^3^. Similarly, moderate heating at 43 °C for a couple of minutes was also found beneficial to enhance pancreatic carcinoma ablation during IRE treatment [35]. Agnass et al. [36], also showed that mild (40 °C) and moderate (46 °C) hyperthermia significantly boost the electroporation effect in microsecond pulse IRE protocols and developed cell viability models that take into account both the electric parameters and temperature increase and could thus predict temperature-dependent cell viability and thermal ablation for pancreatic cancer cells [36].

The cell killing effect of hyperthermia combined with PEF can be explained by the temperature-related increase in tumor tissue conductivity and the concomitant large decrease in tissue impedance [35]. Lower impedance means that the current can more easily flow through the tissue; therefore, the actual value of precipitate energy to the tumor is higher in heated tissues [35]. In addition, hyperthermia increases the fluidity of the cell membrane, resulting in a decrease in the energetic barrier required for electroporation and thus increasing cell death at lower electric field values [36].

The increase in the PRR to 200 Hz (Protocols 2 and 4) resulted in amplified, if not universal, growth inhibition and overall cellular death in pulsed multicellular spheroids exposed to Protocols 2 and 4, respectively. The most pronounced effects (loss of viability in all cells constituting the spheroid and concomitant complete growth inhibition) were observed in Protocol 4, where a train of 2500 pulses of 2 ns duration was applied at PRR of 200 Hz, with an absorbed energy density of 700 mJ/mm^3^ which yielded a temperature elevation (ΔT) of about 22 °C (to reach the final temperature in the bulk of about 45 °C). Similarly, total cellular death and growth inhibition occurred with trains of 500 pulses of 6 and 10 ns as well as 833 pulses of 6 ns, at 200 Hz corresponding to a temperature increase (ΔT) of 32 °C and above (corresponding to a final bulk temperature above 55 °C).

Local or global therapeutic temperature increase is a well-known physical aid that has been exploited in oncology over decades, possibly centuries. Hyperthermia can be applied as an adjuvant treatment, combined with radiotherapy or chemotherapy, and its biological effects are well-known and described elsewhere [37]. In order to determine if hyperthermia alone could be responsible for spheroids growth inhibition and loss of cell viability, we performed the experiments where heated media were applied to multicellular spheroids, in the absence of nsPEF exposure.

The exposure of multicellular spheroids to a peak temperature (of the bulk) of 50 °C resulted in spheroids’ growth inhibition over a period of five days post-exposure. Interestingly, under this condition, the cells did not internalize the propidium iodide, indicating that the membrane remained impermeable and the cells remained viable. At exposures of 60 °C and above, we observed evident thermal damage, which resulted in cell death. These observations indicate that thermal damage probably occurred in nsPEF-exposed spheroids pulsed in HBSS buffer in Protocols 2 and 4 (at pulse durations of 6 and 10 ns), where the final temperatures of the bulk reached 55 °C and more. In contrast, in spheroids exposed to Protocol 4, which involved the application of 2 ns pulses in HBSS buffer, resulting in a bulk temperature of 45 °C, it becomes evident that the temperature rise alone cannot fully explain the complete cell death and that an adjuvant effect occurred between hyperthermia and nsPEFs.

Indeed, the findings of thermal sensitization, highlighted both by others [32,33,34,35,36] and ourselves in the present study, are also in part supported by the observation that when the temperatures of the pulsing buffer do not rise—specifically, when multicellular spheroids were pulsed in ZAP buffer—electric pulses do not affect cellular viability and spheroids’ growth. Nevertheless, other mechanisms could have played a role in cell protection upon nsPEF exposure in ZAP buffer. Importantly, sucrose, which is present in ZAP buffer (and absent in HBSS buffer), is a large solute that reduces cell swelling and could thus have had a beneficial effect [38] in ZAP-exposed spheroids. Moreover, ZAP buffer does not contain calcium ions, while HBSS buffer does, and calcium ions often play a crucial role in nsPEF-induced cell death and/or cell death pathway [39].

In our experimental setting, the role of calcium ions was not studied. It is clear that different cell lines [39] and different clones of the same cell type may have different sensitivity to extracellular Ca^2+^, one example being the EPG85-257 P daunorubicin-sensitive and EPG85-257 RDB -daunorubicin-resistant cells. In experiments using such cell lines, the authors [40] reported that supplementing the sucrose buffer with Ca^2+^ did not significantly reduce the viability of daunorubicin-sensitive cells (while the viability decreased in conductive sucrose-free buffer, regardless of its Ca^2+^ content). In EPG85-257 RDB -daunorubicin-resistant cells, the addition of Ca^2+^ to the sucrose-rich pulsing buffer had no drastic effect on cells, while the addition of Ca^2+^ to the sucrose-free highly conducting buffer decreased the viability from approximately 40% in the medium without Ca^2+^ to approximately 20% in the highly conducting medium supplemented with Ca^2+^. This corroborates the idea that sucrose might have a protective effect.

While Ca^2+^ addition to the sucrose buffer did not drastically decrease viability in some experiments [40], another study indicates that the presence of calcium ions in a sucrose-containing solution can markedly decrease cell survival by about five-fold in 1.5 and 5 h after exposure to nsPEF [41]. As nicely demonstrated by Pakhomova and colleagues [41], pore formation, cell survival time, and cell death type are determined by the level of external Ca^2+^ concentration during but also after the exposure of 2D-grown cells to nsPEF. Precisely, when increasing the Ca^2+^ level to 2 mM post-nsPEF exposure, necrotic death could be observed 60 to 90 min after PEF exposure [41].

## 4. Materials and Methods

### 4.1. 3D Multicellular Spheroids

Human colorectal carcinoma cells HCT-116 (ATCC^®^ CCL-247TM) stably expressing green fluorescent protein (GFP) [42] were cultured in Dulbecco’s Modified Eagle Medium (DMEM, ThermoFisher Scientific, Illkirch, France) containing 1.8 mM calcium chloride, 2.4 × 10^−4^ mM ferric nitrate, 0.81 mM magnesium sulfate, 5.33 mM potassium chloride, 44.0 mM sodium bicarbonate, 110.3 mM sodium chloride, 0.9 mM sodium phosphate monobasic, 25 mM D-Glucose (Dextrose), 0.03 mM phenol red, 1 mM sodium pyruvate, 4.5 g/L of glucose (Gibco-Invitrogen, Carlsbad, CA, USA), L-Glutamine (CSTGLU00, Eurobio, France) and pyruvate, supplemented with 10% of fetal bovine serum (F7524, Sigma, Waltham, MA, USA), and 1% of penicillin/streptomycin (P0781, Sigma, USA). Cells were cultured in a humidified atmosphere at 37 °C supplemented with 5% of CO_2_. Cells were tested negative for mycoplasma with MycoAlert Mycoplasma Detection kit (cat n°#LT07-318, Lonza, Basel, Switzerland). To generate multicellular spheroids, 500 cells were suspended in 200 µL of culture medium and seeded in Costar^®^ Corning^®^ Ultra-low attachment 96-well plates (Fisher Scientific, Illkirch, France). Spheroids were grown in a humidified atmosphere at 37 °C with 5% CO_2_. Cell aggregation occurred within the first 72 h following the seeding and allowed for the formation of single spheroids of similar sizes in each well [26]. Five-day-old spheroids were exposed to nsPEF in 0.5 mL of pulsing buffer (as described below), and ten minutes after the pulse, the spheroids were transferred from the pulsing medium into the medium containing 1 µM of PI and monitored for the following 5 days (Figure 8). Precisely, the spheroids were taken out of the pulsing medium with a pipette, and after a couple of seconds, when the spheroids reached the bottom of the pipette cone, we transferred the spheroids by gently touching the spheroids-containing cone with the culture medium containing PI in individual well of the multi well plate.

### 4.2. nsPEF Generator

Figure 8A shows the schematic representation of the specific experimental setup used for spheroids electropulsation. A commercial generator (FPG 20-1NJ10, FID Technology, Burbach, Germany) with a 50 Ω output impedance was used for the generation of unipolar pulses of approximately 2, 4, 6, 8, and 10 ns (±0.3 ns), as shown in Figure 8B (bottom panel). Pulses were delivered to cells using a pair of stainless-steel electrodes with a 1 mm gap precisely set using a micromanipulator on the bottom of a plastic Petri dish where a single cellular spheroid was located during exposure. Pulse shapes and amplitudes were continuously monitored with a 1 GHz oscilloscope (DPO 4104, Tektronix, Beaverton, OR, USA) connected between the pulse generator and the electrodes through a tap-off box (Model 245-NMFFP-100, Barth Electronics, Inc., Boulder, NV, USA). The latter is a three-port device with a high measurement port impedance of 4950 Ω corresponding to an attenuation of 40 dB with additional 30 dB attenuators that allow the real-time monitoring of pulses arriving at the load, namely, the spheroids.

### 4.3. Pulsing Buffers

Two types of buffers were used for spheroids exposure, including Hanks’ Balanced Salt Solution (HBSS, Gibco, containing 1.26 mM calcium chloride, 0.49 mM magnesium chloride, 0.41 mM magnesium sulfate, 5.33 mM potassium chloride, 0.44 mM potassium phosphate monobasic, 4.17 mM sodium bicarbonate, 137.9 mM sodium chloride, 0.34 mM sodium phosphate dibasic anhydrous, and 5.55 mM D-Glucose (Dextrose)) and a low-conductivity iso-osmotic pulsation buffer, also known as “ZAP” buffer (containing 8.1 mM dipotassium phosphate, 1.9 mM monopotassium phosphate, 1 mM magnesium chloride, and 250 mM sucrose in water; pH: 7.4, osmolarity: 270 mOsm/L). Electrical conductivity (*σ*) was measured using a slim dielectric probe (85070E Dielectric probe kit, Agilent, Santa Clara, CA, USA), and it was equal to 0.2 and 1.4 S/m for ZAP and HBSS buffers, respectively, for frequencies lower than a few hundred MHz. Because of the different pulsing buffer conductivity, to ensure the correct match of the load to the generator impedance (50 Ω), an open circuit or a 50 Ω charge were set in parallel with the electrodes when using HBSS or ZAP buffers, respectively. This resulted in almost identical pulse shapes delivered to the spheroids, with differences within 10% (Figure 9). Applied pulses are obtained by summing the incident and the reflected pulses after time compensation for the offset due to the delay between the forward and reflected pulses [43]. Specifically, using 10 kV bias voltage, the peak amplitude of the electric field measured in HBSS was equal to about 10 kV (green curve), while the amplitude of the electric field in the ZAP buffer was equal to approximately 9 kV (red curve).

### 4.4. Exposure Protocols

#### 4.4.1. Pulsed Electric Field Protocols

Four exposure protocols, as summarized in Table 1, were tested in both HBSS and ZAP pulsing buffers to explore the potential effects of different pulse parameters, including (1) pulse duration (2–10 ns), (2) repetition rate (20 or 200 Hz), (3) cumulative time duration (1–5 µs), and (4) absorbed electrical energy density (up to 81 mJ/mm^3^ in ZAP and up to 700 mJ/mm^3^ in HBSS). Per each experimental protocol, the cumulative time duration (D, [μs]) was calculated as the duration of the pulse (d) multiplied by the number of pulses (N), while the absorbed electrical energy density (u [J/m^3^]) was calculated using [44]:u = σ × E^2^ × N × d(1)
with σ the electrical conductivity of the pulsing buffer (S/m), E the electric field amplitude (V/m), N the number of pulses, and d (ns) the pulse duration.

The minimum number of pulses of 500 was chosen as a reference in Protocols 1 and 2 to establish the effects induced when varying the pulse length.

In Equation (1), the current density is equal to σ × E (A/m^2^), which can then be multiplied by E (V/m) to obtain an absorbed power density (V × A/m^3^). The resistance of the electrodes containing the buffer is around 50 ohms. Using the relation between resistance and applied voltage (9 kV and 10 kV in ZAP and HBSS, respectively), a current of 180 and 200 A is obtained in low and high conducitivy buffer, respectively. The applied energy is obtained by multiplying the voltage, current, and cumulative time duration (D). The applied energy is comprised between 1.6 and 8 J (in ZAP) and 14 and 70 J (in HBSS) for 1 and 5 μs, respectively.

#### 4.4.2. Hyperthermia Protocols

In order to evaluate the impact of elevated temperatures on spheroids viability and growth (positive control—hyperthermia), we conducted an experiment where multicellular spheroids were exposed to a cellular medium that had been heated to temperatures of 50 °C, 60 °C, 70 °C, or 75 °C.

The procedure involved gently applying 1 mL of heated medium onto a 20-µL droplet of medium (at 23 °C) that contained six multicellular spheroids. This mixture of spheroids and medium was then allowed to gradually cool down to 23 °C. Throughout this cooling process, which took ≤2 min, the temperature reduction was monitored using a temperature fiber-optic probe.

Once the temperature had reached a stable level, individual spheroids were carefully transferred to separate wells for further analysis and monitored for a period of 5 days using the procedure outlined in Section 4.6.

### 4.5. Electromagnetic and Thermal Dosimetry

#### 4.5.1. Electromagnetic Dosimetry 

Numerical simulations of the electric field distribution between the electrodes at the location of the exposed spheroid have been performed in detail in [26]. Briefly, the Finite Difference Time Domain (FDTD) numerical method was used to calculate the electric field distribution both inside and outside the spheroid. Under the current exposure conditions, i.e., a 1 mm gap between the electrodes and an electric field peak amplitude of 9 kV and 10 kV, each spheroid was approximately exposed to a rather homogeneous field of 90 kV/cm and 100 kV/cm, in ZAP or HBSS, respectively.

#### 4.5.2. Thermal Dosimetry 

Local temperature elevation, within an approximated volume of 0.5 mm^3^, induced in ZAP and HBSS buffers for the 4 protocols was measured using a fiber-optic probe (Luxtron One; Lumasense Technologies, Santa Clara, CA, USA) located between the electrodes where multicellular spheroids were set during exposure. Temperature measurements were performed without cells and repeated at least 3 times per condition explored. Thermal dosimetry was carried out under conditions identical to those for exposures of the spheroids to nsPEF. The electrodes have a length of 6 mm in contact with the buffer, a height of 0.6 mm, and a width of 1.8 mm. The gap between the two electrodes is 1 mm. The quantity of buffer solution represents a volume of 500 µL. The spheroids have a diameter of around 0.5 mm, corresponding to a volume of 0.06 µL. The temperature probe is an optical fiber coated in a dielectric (PTFE) with a total diameter of 0.6 mm. If we consider the total height of the buffer, the volume occupied at the level of the electrodes by the probe is 0.2 µL. To compare the volumes occupied by the spheroid and the temperature probe, we consider only the volume between the electrodes where the heating primarily occurs (6 × 0.6 × 1 mm^3^), i.e., 3.6 µL. As illustrated in Figure 10, volume ratios between spheroids or probes and the heated volume are small, namely around 2% (0.06/3.6 µL) and 6% (0.2/3.6 µL) for the spheroids and the probe, respectively.

The thermal probe is immune to radiofrequency (RF) interference, electrically non-conductive, and minimally heat-conductive. The used probe is a fast-response immersion probe with a response time of 250 ms in water and is well adapted for measuring a small sample that is being RF heated. For a single channel configuration, the measurement rate is 4 Hz maximum, which is in line with the response time. It is adapted for temperature measurements between 0 and 295 °C. Due to the small size of the probe compared to the heating volume, the probe has a limited impact on heat dissipation into solution convection obstruction. However, due to the localized heated area, there is thermal convection and diffusion in the whole buffer, as well as exchanges with the air surrounding the buffer. The spheroid represents a very small part of the pulsing buffer. Thus, the proposed measurements allow for an assessment of the temperature in the environment where the spheroids are exposed to nsPEF.

### 4.6. Microscopy 

To evaluate the effects of nsPEF exposure or hyperthermia exposure (considered a positive control), each spheroid was individually placed in a well of the Costar^®^ Corning^®^ Ultra-low attachment 96-well plates (Fisher Scientific, Illkirch, Alsace, France) containing culture medium. Spheroids were subsequently followed over a period of 5 days after nsPEF exposure using bright field and fluorescence microscopy, performed over time with the IncuCyte Live Cell Analysis System Microscope (Sartorius, Epsom, UK) using a magx10 objective. Propidium iodide (1 µM PI) was added to the culture medium 10 min post-nsPEF or heated medium exposure to assess cell viability in real-time, and the red fluorescence due to PI penetration into the spheroids was followed for 32 h. The green fluorescence was exploited to determine the spheroid’s growth over time and was monitored for a period of 5 days post-nsPEF exposure. Single-plane wide-field micrographs were acquired. The focal plane was centered at the equatorial position of the spheroid. The size of the image was 1392 × 1040 pixels (1.22 µm/pixel), and the acquisition time was set to 400 ms and 800 ms for green and red fluorescence channels, respectively. Images were analyzed with the ImageJ software 1.53 (U.S. National Institute of Health, Bethesda, MD, USA). The software was used to determine the mean fluorescence intensity of spheroids and to measure the spheroids’ area in the equatorial plane.

### 4.7. Statistical Analysis 

Statistical analyses were performed using Prism 7 software and analyzed by a two-way ANOVA. Overall statistical significance was set at *p* < 0.05. Data were expressed as mean ± standard error of the mean (SEM) or mean ± standard deviation (SD) (as indicated in dedicated sections). At least 6 biological replicates (n ≥ 6 spheroids) were exposed individually in one or more independent experiments (N), as indicated in dedicated sections in figure captions, presenting viability and growth data.

## 5. Conclusions

To conclude, our study indicates that the efficacy of nsPEFs with 500 to 2500 pulses of 2 to 10 ns of duration and electric field intensity of about 100 kV/cm, applied at PRR of 20 and 200 Hz to ablate 3D-grown cancer cells, highly depends on the medium in which the cells are exposed to electric pulses. While highly conductive Ca^2+^-rich medium (HBSS), under specific pulsing conditions (namely the PRR of 200 or 20 Hz), may or may not yield cell death at given pulse parameters, delivered in a low-conductivity medium, the pulses do not result in cell ablation. The presented experiments do not allow us to draw a conclusion regarding whether conductance alone plays a role in the ablation of multicellular spheroids upon nsPEF exposure in HBSS buffer. This is because the presence of calcium ions or the uptake of water during PEF exposure and the consequent cell swelling may have led to cell membrane rupture, whereas this phenomenon was less likely to occur in the ZAP buffer. Nevertheless, our results show that when the high-conductivity HBSS medium is used and high PRRs are applied, pulsing buffer temperatures increase, and when the temperatures attain values above 50 °C, thermal damage occurs. When a sweet spot is reached, the locally induced mild hyperthermia, which by itself is not lethal to cells (in our case 45 °C), enhances the effect of PEFs and results in efficient cell kill, even when pulse durations as low as 2 ns are applied. This indicates that a local temperature rise increases cellular sensitivity and boosts nsPEF effects to induce cell death. This finding suggests that cell sensitization in electroporation protocols might also be beneficial from an engineering point of view to enhance the deleterious effects on cells while maintaining a minimal electric field amplitude. However, as local (mild) temperature rise can be beneficial and might result in more efficient cell kill with nsPEFs, caution should be applied when nsPEFs of short duration will be translated in vivo, where local heat dissipation will be more important due to blood circulation and local cooling, which could potentially result in lesser cell sensitization and consequently less efficient cell kill upon short nsPEF exposure. Altogether, our results show that depending on the pulse parameters employed (including the medium in which the cells are exposed to electric pulses) and corresponding induced temperature elevation, two main pathways leading to cellular death and growth inhibition can be distinguished, mainly depending on the PRR. One is electroporation, and the other is thermal ablation. While our results strongly suggest that increased tissue heating enhances the killing effect of nsPEF, the protective role of the pulsing buffer (or, more broadely, the extracellular environment) clearly has to be investigated in further studies.

## Figures and Tables

**Figure 1 ijms-24-14999-f001:**
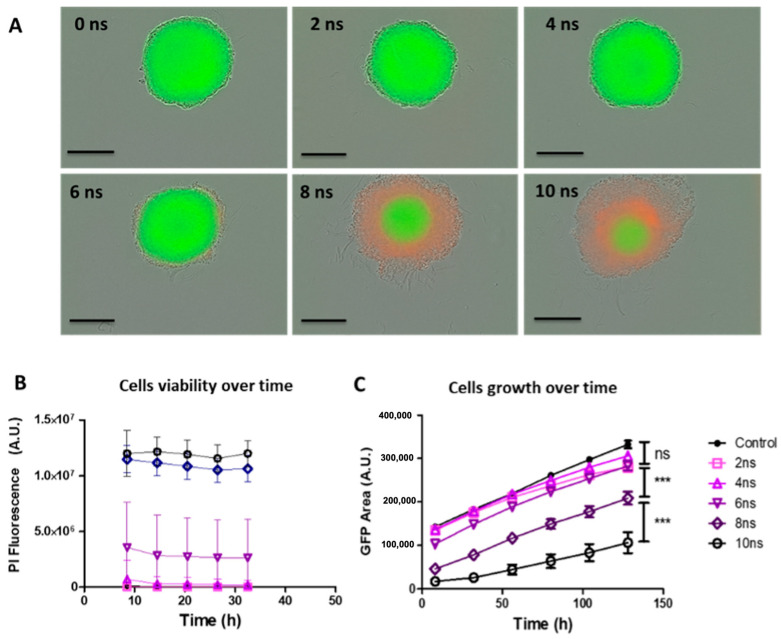
Cell viability and growth of multicellular spheroids depend on the pulse duration at 20 Hz. HCT 116-GFP spheroids were treated with 500 unipolar pulses lasting 0 (Control), 2, 4, 6, 8, and 10 nanoseconds (ns) delivered at a pulse repetition rate of 20 Hz with a 100 kV/cm electric field. After exposure, spheroids were incubated in culture medium containing 1 µM PI during wide-field fluorescence video microscopy over a period of 5 days. (**A**) Representative micrographs of spheroids with dead cells, which internalized propidium iodide (PI) exhibiting a red fluorescence and viable cells, constitutively expressing the green fluorescence protein (GFP) exhibiting the green fluorescence 24 h post nPEF exposure. Scale bar: 300 µm. (**B**) Cell death reflected by PI uptake mean fluorescence intensities (A.U.) plotted as a function of time (h). N = 1 experiment, n = 6 spheroids per condition. (**C**) Growth curves plotted from GFP fluorescence area (A.U.) as a function of time (h). N = 3 experiments, n = 6 spheroids per condition. Data are reported as mean ± standard error mean. Two-way ANOVA *** *p* < 0.001.

**Figure 2 ijms-24-14999-f002:**
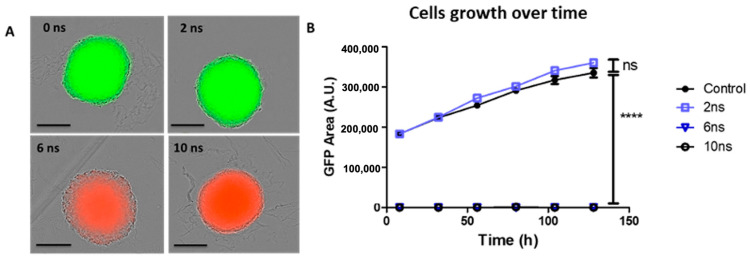
Spheroid cell viability and growth depend on the pulse duration at 200 Hz. HCT 116-GFP spheroids were treated with 500 unipolar pulses lasting 0 (Control), 2, 6, and 10 nanoseconds (ns) at 200 Hz with a 100 kV/cm electric field intensity. After exposure, spheroids were incubated in culture medium containing 1 µM PI during wide-field fluorescence video microscopy over a period of 5 days. (**A**) Representative micrographs of spheroids highlighting non-viable cells, which internalized propidium iodide (PI) exhibiting a red fluorescence and viable cells, constitutively expressing the green fluorescence protein (GFP) exhibiting the green fluorescence 24 h post nPEF exposure. Scale bar: 300 µm. (**B**) Growth curves plotted from GFP fluorescence area (A.U.) as a function of time (h). N = 2 experiments, n = 11 spheroids for control and 10 ns and n = 6 spheroids for 2 ns and 6 ns. Data are reported as mean ± standard error mean. Two-way ANOVA **** *p* < 0.0001.

**Figure 3 ijms-24-14999-f003:**
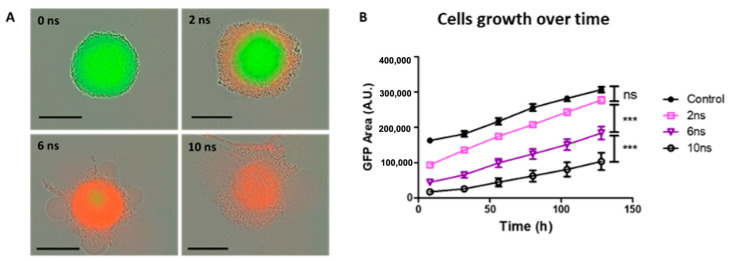
Spheroid cell viability and growth depend on the pulse duration for constant cumulative time duration and absorbed energy density at 20 Hz. HCT 116-GFP spheroids were treated with 0 (Control), 2500, 833, and 500 unipolar pulses lasting 0, 2, 6, and 10 nanoseconds (ns), respectively, at 20 Hz with 100 kV/cm electric field. After exposure, spheroids were incubated in culture medium containing 1 µM PI during wide-field fluorescence video microscopy over a period of 5 days. (**A**) Representative images of spheroids with dead cells, which internalized propidium iodide (PI), exhibiting a red fluorescence, and viable cells, constitutively expressing the green fluorescence protein (GFP) 24 h post-nPEF exposure. Scale bar: 300 µm. (**B**) Growth curves plotted from GFP fluorescence area (A.U.) as a function of time (h). Data are reported as mean ± standard error mean. N = 3 experiments, n = 6 spheroids per condition. Two-way ANOVA *** *p* < 0.001.

**Figure 4 ijms-24-14999-f004:**
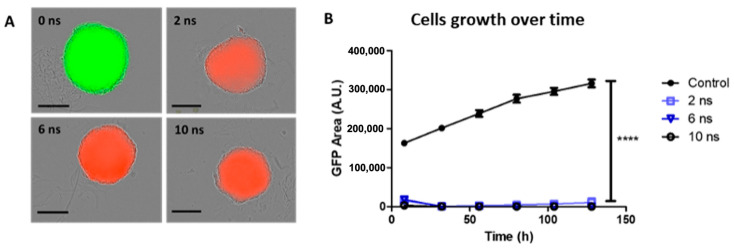
Spheroid cell viability and growth inhibition depend on the pulse duration for constant cumulative time duration and absorbed energy density at 200 Hz. HCT 116-GFP spheroids were treated with 0 (Control), 2500, 833, and 500 unipolar pulses lasting 0, 2, 6, and 10 nanoseconds (ns), respectively, at 200 Hz with 100 kV/cm electric field intensity. After exposure, spheroids were incubated in culture medium containing 1 µM propidium iodide (PI) during wide-field fluorescence video microscopy over a period of five days. (**A**) Representative images of spheroids with dead cells, which internalized propidium iodide (PI) exhibiting a red fluorescence and viable cells, constitutively expressing the green fluorescence protein (GFP) 24 h post nPEF exposure. Scale: 300 µm. (**B**) Growth curves plotted from GFP fluorescence area (A.U.) as a function of time (h). Data are reported as mean ± standard error mean. N = 2 experiments, n = 6 spheroids per condition. Two-way ANOVA **** *p* < 0.0001.

**Figure 5 ijms-24-14999-f005:**
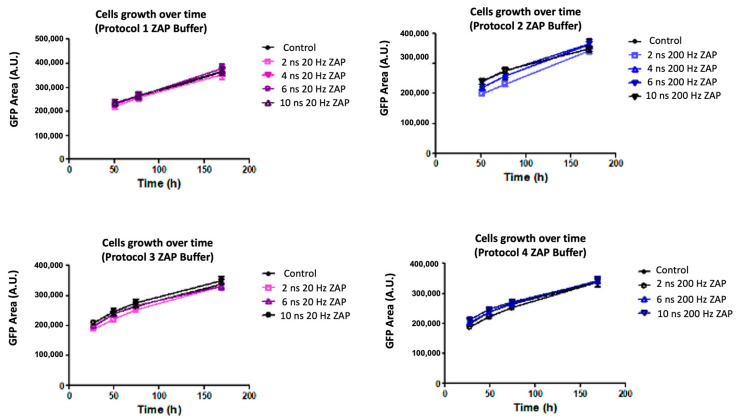
Spheroids growth in low conductivity buffer in Protocols 1, 2, 3, and 4. HCT 116-GFP spheroids were treated by Protocols 1, 2, 3, and 4 in a low-conductivity medium (ZAP) with a 90 kV/cm electric field. After exposure, spheroids were incubated in culture medium containing 1 µM propidium iodide (PI) during wide-field fluorescence video microscopy over a period of 5 days. Growth curves are plotted from green fluorescence protein (GFP) fluorescence area (A.U.) as a function of time (h). Data are reported as mean ± standard error mean. N = 1 experiments, n = 6 spheroids per condition.

**Figure 6 ijms-24-14999-f006:**
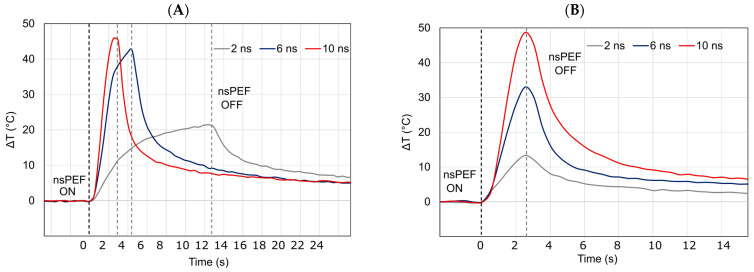
nsPEF induced temperature elevation in high-conductivity buffer (HBSS). Representative examples of the nsPEF-induced temperature elevation in HBSS following the exposure to 2, 6, and 10 nsPEF at 100 kV/cm in (**A**) Protocol 4, i.e., constant cumulative time duration at 200 Hz, and (**B**) Protocol 2, i.e., 500 pulses at 200 Hz with increasing time duration. Black and gray dashed lines represent the beginning and the end of the nsPEF exposure, respectively. The temperature elevation graphs not only display the local temperature increase values, but also emphasize the short duration of the temperature spikes.

**Figure 7 ijms-24-14999-f007:**
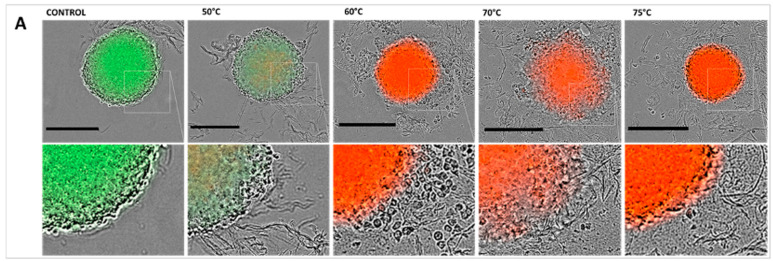
Effect of hyperthermia on spheroids viability and growth. HCT-116 GFP spheroids exposed to heated cellular medium. After exposure, the spheroids were incubated in culture medium containing 1 µM propidium iodide (PI) during wide-field fluorescence video microscopy over a period of five days. (**A**) Representative micrographs of spheroids with viable cells, constitutively expressing the green fluorescence protein (GFP) exhibiting the green fluorescence 24 h post-heated medium exposure, and dead cells, which internalized PI exhibiting red fluorescence. Scale bar: 300 µm. (**B**) Growth curves plotted from GFP fluorescence area (µm^2^) plotted as a function of time (days). (**C**) Growth curves plotted from the bright field image of the spheroid’s area (µm^2^) plotted as a function of time (days). (**D**) Cell viability reflected by PI uptake mean fluorescence intensities (A.U.) plotted as a function of time (days). N = 1 experiment, n = 6 spheroids per condition. Data are reported as mean ± standard deviation.

**Figure 8 ijms-24-14999-f008:**
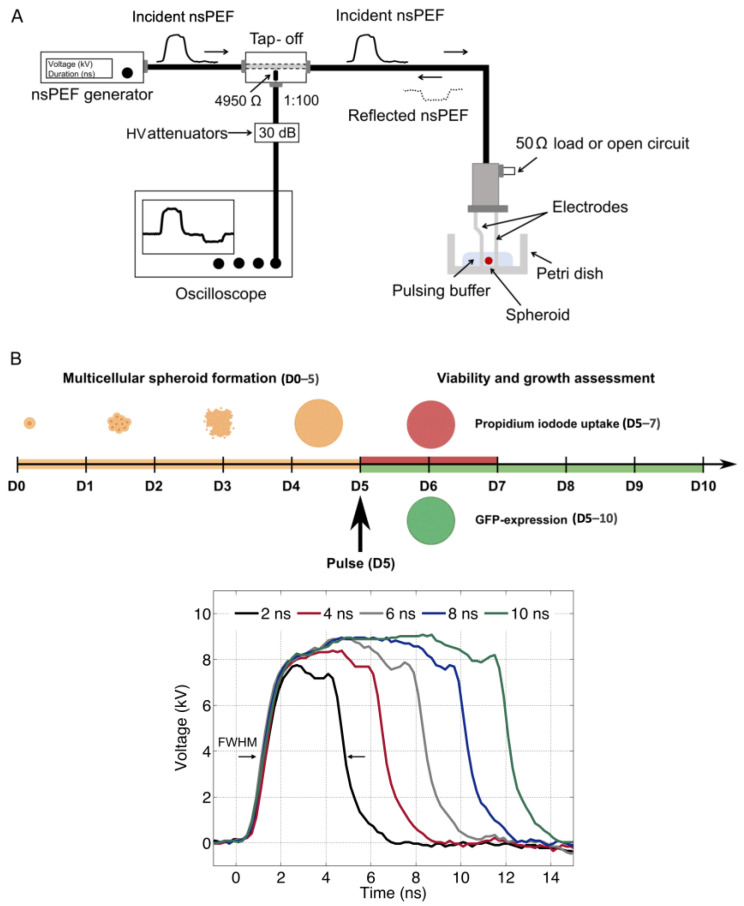
(**A**) Schematic representation of the experimental setup composed of a unipolar nsPEF generator, a tap-off, an attenuation chain, an oscilloscope, a pair of electrodes, and the Petri dish containing a single spheroid. (**B**) Experimental protocol (upper panel) with the timeline of multicellular spheroid formation (D0–5), pulse generation and delivery (D5), viability (D5–7), and growth assessment (D5–10). Unipolar voltage pulses (lower panel) obtained using a bias voltage of 10 kV with Full Width at Half Maximum (FWHM) of approximately 2 ns, 4 ns, 6 ns, 8 ns, and 10 ns.

**Figure 9 ijms-24-14999-f009:**
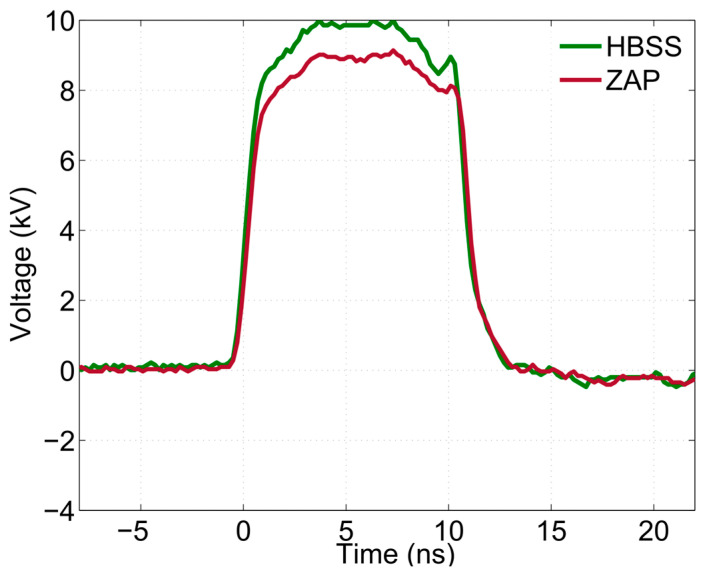
10 nsPEF delivered in different pulsing buffers. Unipolar voltage pulses in HBSS (green curve) and ZAP (red curve) media obtained using a bias voltage of 10 kV.

**Figure 10 ijms-24-14999-f010:**
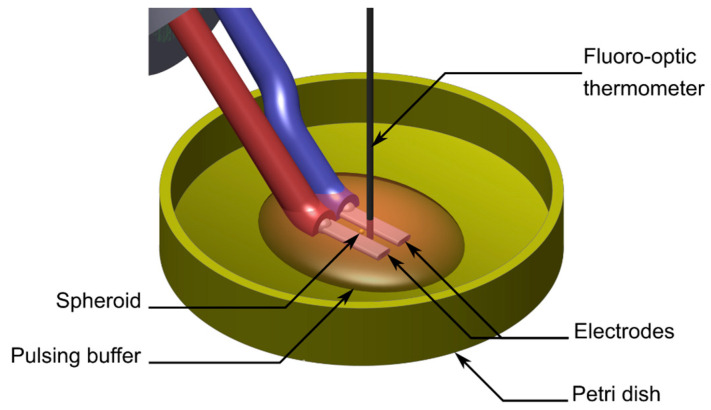
3D representation of the experimental set-up used to perform temperature measurements with a fiber-optic thermometer located between the electrodes where the multicellular spheroid was located during exposure.

**Table 1 ijms-24-14999-t001:** Summary of pulses parameters in examined exposure protocols and measured temperature increase (ΔT). PRR = pulse repetition rate (Hz); NA = not applicable, as the temperature increase was insignificant (below 1 °C). Number of independent experiments (N) in ZAP was equal to 1. For all experiments number of biological replicates (n) was ≥6.

	Pulse Duration d (ns)	Number of Pulses N	Cumulative Time Duration D (μs)	Total Exposure Duration t (s)	Absorbed Energy Density u (mJ/mm^3^) in HBSS	Temperature Increase-ΔT (°C) in HBSS	Absorbed Energy Density u (mJ/mm^3^) in ZAP	Temperature Increase-ΔT (°C) in ZAP	E-Field Intensity in HBSS (kV/cm)	E-field Intensity in ZAP (kV/cm)	Number of Independent Experiments HBSS (N)
Protocol 1											
(PRR = 20 Hz)	2	500	1	25	140	3.8 ± 0.2	16	NA	100	90	3
	4	500	2	25	280	5.7 ± 0.3	32	NA	100	90	3
	6	500	3	25	420	9.1 ± 1.8	47	NA	100	90	3
	8	500	4	25	560	10.8 ± 2.2	65	NA	100	90	3
	10	500	5	25	700	15.1± 2.7	81	1.1 ± 0.2	100	90	3
Protocol 2											
(PRR = 200 Hz)	2	500	1	2.5	140	12.6 ± 0.8	16	1.2 ± 0.1	100	90	2
	6	500	3	2.5	420	32.2 ± 0.7	47	2.5 ± 0.1	100	90	2
	10	500	5	2.5	700	46.8 ± 3.9	81	4.4 ± 0.2	100	90	2
Protocol 3											
(PRR = 20 Hz)	2	2500	5	125	700	8.5 ± 1.5	81	NA	100	90	3
	6	833	5	41.6	700	15.2 ± 1.9	81	NA	100	90	3
	10	500	5	25	700	16.8 ± 2.2	81	1.1 ± 0.2	100	90	3
Protocol 4											
(PRR = 200 Hz)	2	2500	5	12.5	700	22.0 ± 2.2	81	3.2 ± 0.1	100	90	2
	6	833	5	4.16	700	42.8 ± 1.4	81	3.7 ± 0.4	100	90	2
	10	500	5	2.5	700	46.8 ± 3.9	81	4.4 ± 0.2	100	90	2

## Data Availability

Data available for a special request of an interested reader (contact with corresponding author).

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
