# Peer review of "Effects of Nanosecond Pulsed Electric Field (nsPEF) on a Multicellular Spheroid Tumor Model: Influence of Pulse Duration, Pulse Repetition Rate, Absorbed Energy, and Temperature"

_ijms, 2023, doi:10.3390/ijms241914999_

Round 1

Reviewer 1 Report

This manuscript by Orlacchio, et al. describes the effects of 2-10 ns-long pulses on multicellular spheroid growth and death. GFP is used to monitor growth and PI is used to label dead cells. There are not many previous studies using pulses in this 2-10 ns range and even fewer looking at the effect on spheroids, so these observations are sorely needed. However, the data provided is missing the key element of current density applied, making it impossible to calculate the energy delivered. Hopefully, those data are available, and their addition will make this study more useful. There are several detailed suggestions for improvement below and I look forward to a revised version of this manuscript.

line

Title:      Much of this paper is a discussion of heat effects on the spheroids so I would recommend changing the title by replacing “buffer properties” with “temperature”

37           Ref 5:    better to use: Nuccitelli, R. (2016). Tissue Ablation Using Nanosecond Electric Pulses. In: Miklavcic, D. (eds) Handbook of Electroporation. Springer, Cham. https://doi.org/10.1007/978-3-319-26779-1_93-1

48           A better reference to the difference in muscle contraction is Gudvangen, E., Kim, V., Novickij, V., Battista, F. & Pakhomov, A.G. Electroporation and cell killing by milli- to nanosecond pulses and avoiding neuromuscular stimulation in cancer ablation. Scientific reports 12, 1763 (2022).

59           When discussing cellular effects of imposed fields on spheroids that are close to a mm in diameter, it is more appropriate to use kV/cm than MV/m to describe the imposed field

94           We have known for decades that the effects of ns pulses depend on pulse width and the shorter the pulse the more pulses are needed to have similar effect. The energy delivered is proporational to pulse width. So I am surprised that you used the same pulse number for each of these pulse widths. You should explain your rationale here as it is not clear.

104        I suspect that you could get spheroid death with the shorter pulses if you used more of them so not sure what this tells you. It would be better to determine how many 2ns pulses are needed to observe PI uptake and GFP shrinkage.

Fig. 3A  10ns shows no GFP left, but the graph in fig. 3B does not indicate that. Please explain that discrepancy

151        In order to calculate mJ/mm3, current density is needed but you never say what this was. Please include those data.

186        Where is Table I. You should move it to better location so reader can see it when needed. Also, it would be good to add E field applied and current density for each protocol and pulse duration.

188        I suspected hyperthermia would be involved based on 200Hz but could not calculate the expected increase in temperature without the current density. You need to include those data and do the calculation of expected temperature increase at this point in the paper.

275        “…electric pulses of 100-200ns

279        Better reference for pulses targeting mitochondria: Nuccitelli, et al. Nano-Pulse Stimulation Induces Changes in the Intracellular Organelles in Rat Liver Tumors Treated In Situ. Lasers Surg Med 52, 882-889 (2020).

Author Response

Please find attached the file with the point-by-point replies to your comments, 

best regards 

on the behalf of the authors, 

Rosa Orlacchio  

Reviewer 2 Report

This is a straightforward study and the reported results are what one would expect. It comes without doubt that higher doses of nsPEF and more heating will cause more cell death. The paper could benefit from testing protocols designed to find answers which are not obvious, e.g., whether pulse duration per se is essential for cell killing (use different pulse durations and tune PRR to deliver the same dose in the same time interval) and from testing differently formulated solutions (see below).

The paper concludes that lower cell death in low-conductivity solution is caused by reduced heating. This can be just partially true. The low-conductivity medium has no Ca2+ (which is a major factor that determines cell survival after electroporation) and its major component is sucrose (which is a large molecule that prevents cell death from swelling). These well-established cell killing mechanisms after nsPEF have not been considered. It looks imperative to amend the study by experiments where a low conductance solution is supplemented with Ca2+ and contains a smaller principal solute.

A major problem that potentially undermines the entire study is the accuracy of temperature measurements. The authors are surprisingly brief in their description of thermometry, which is a challenge when we measure heating in a volume as small as 0.5 mm3. The mere presence of the fiber optic probe (with the size comparable to the heated volume) can have multiple effects with  unpredictable impact: it can be a heat sink itself, it can prevent heat dissipation into solution, in can obstruct convection, it will not equalize with the sample due to its large heat capacity, and, being made of a dielectric material, it will reduce heating compared to a situation when the same volume is occupied by the conductive solution (and this error will be different for low- and high-conductivity solutions). On top of it, measurements were performed in the absence of spheroids; it is more likely than not that heating in the presence of spheroids and within spheroids is significantly different from what is measured in solution (even if solution measurements are fully accurate). The authors must convince readers that temperature measurements were accurate, and the impact of all the factors listed above was properly taken into account or proven negligible. This is mandatory.

Minor points:

Please add the exact time when pulsing media was replaced by culture media, and how. Was it in 10 min when PI was added?

Please use standard terminology when talking about “dose”

There is nothing to prove “synergistic” effect. Maybe it was additive? You just did not study it.

Please add HBSS composition..

Table 1: Please add the number of experiments for each condition.

How comes that two last treatments in Protocol 3 caused the same heating, although one treatment took almost twofold longer? Please explain.

some grammar editing is needed

Author Response

Please find attached the file with the point-by-point replies to the reviewer, 

on the behalf of the authors, 

best regards, 

Rosa Orlacchio 

Round 2

Reviewer 1 Report

This revised version of the manuscript has addressed all of my concerns and is now ready for publication.

Author Response

We thank again the Reviewer for his / her constructive feedback, 

on the behalf of the authors, 

Rosa Orlacchio  

Reviewer 2 Report

The authors have made a genuine effort to address the criticisms. I appreciate the effort and agree with the amendments, with one exception.

The exception relates to the role of Ca and sucrose in cell death. The long paragraph added to the discussion (should be shortened) does not disprove the likely role of Ca ions in the lethal effects of nsPEF. Many studies by independent groups (Craviso, Ibey, Pakhomov, Pakhomova, Vernier) showed that Ca or YO-PRO-1 uptake is by far the strongest during the first 1-2 min after electroporation by nsPEF and tends to come to plateau afterwards.  Changing solution as late as in 10 min after electroporation could allow sufficient time for Ca to enter cells and exert its toxicity. In the properly cited study by Pakhomova and colleagues [40], solution was changes as early as in 20-30 s after electroporation, and this is a big difference compared to 10 min.

The same study also demonstrates that the presence of Ca in a high sucrose solution can markedly decrease survival, about 5-fold in 1.5 and 5 hr after exposure (at the alter 24-hr time point, survival was low in all groups).

Furthermore, the discussion focuses entirely on Ca, but does not discuss another point that sucrose is a large solute that prevents cell swelling and may also be protective (see Pakhomova et al., PLoS ONE 8(7): e70278). It is feasible to expect that smaller solutes like adonitol (Nesin et al., Biochimica et Biophysica Acta 1808 (2011) 792–801) will be less protective despite the same conductance, although this reviewer does not know if this was experimentally proven.

To make it clear: Maybe it was indeed the impact of conductance only (as the paper currently suggests), but maybe it was the impact of Ca or of sucrose, or a combination of them. The experiments performed are not sufficient to make an unequivocal conclusion, and published studies do not give this answer either. If the authors do not plan to perform additional experiments, conclusions must be carefully moderated, pointing to all possibilities, with proper references and balanced discussion. The language in the abstract needs to be adjusted as well.

Minor points:

1.      The word “synergistic” was replaced with “additive”. Sorry for misunderstanding, I did not suggest “additive”, it was just an example; you did not study it either. Just say that heating enhanced the killing effect of nsPEF.

2.      Indicating the number of experiments as “1” in the table looks strange. Further in Methods it looks like it was actually six rather than one, maybe you should change it. What is “biological replicate”?

3.      “osmolarity: 270 osmol/L”   It is mOsm/L 

4.      For lines 320-330 and elsewhere, please update the language to make clear if you are talking about the change in temperature (delta t) of final temperature reached. The text is a bit confusing now.

needs some improvement

Author Response

Please, find our replies in the file attached, 

On the behalf of the authors, 

Rosa Orlacchio 
